# Polyphenols from Olive-Mill Wastewater and Biological Activity: Focus on Irritable Bowel Syndrome [note 1]

**DOI:** 10.3390/nu14061264

**Published:** 2022-03-16

**Authors:** Francesca Curci, Filomena Corbo, Maria Lisa Clodoveo, Lara Salvagno, Antonio Rosato, Ivan Corazza, Roberta Budriesi, Matteo Micucci, Laura Beatrice Mattioli

**Affiliations:** 1Department of Pharmacy-Drug Sciences, University of Bari “Aldo Moro”, 70125 Bari, Italy; francesca.curci@uniba.it (F.C.); filomena.corbo@uniba.it (F.C.); lara.salvagno@uniba.it (L.S.); antonio.rosato@uniba.it (A.R.); 2Interdisciplinary Department of Medicine, School of Medicine, University of Bari “Aldo Moro”, 70124 Bari, Italy; marialisa.clodoveo@uniba.it; 3Department of Experimental Diagnostic and Specialty Medicine (DIMES), Alma Mater Studiorum University of Bologna, 40138 Bologna, Italy; ivan.corazza@unibo.it; 4Department of Pharmacy and Biotechnology, Food Chemistry and Nutraceutical Lab, Alma Mater Studiorum-University of Bologna, 40126 Bologna, Italy; laura.mattioli13@unibo.it; 5Department of Biomolecular Sciences, University of Urbino “Carlo Bo”, 61029 Urbino, Italy; matteo.micucci@uniurb.it; 6UniCamillus-Saint Camillus International University of Health Sciences, Via di Sant’Alessandro, 800131 Rome, Italy

**Keywords:** irritable bowel syndrome, ileum and colon contractility, antimicrobial activity, MOMAST plus30

## Abstract

Waste represents a cost for companies, in particular for agro-food companies, which can become a resource as a secondary material. In this work, we examine three products of olive-oil waste water, named **MOMAST^®^** (**Plus30**, **PW25**, and **HY100**). Based on the chemical composition, obtained with different methods, we hypothesized a possible application as food supplements in irritable bowel syndrome (IBS). We therefore studied **MOMASTs** on some targets linked to this pathology: antioxidant action and spontaneous and induced intestinal contractility of the ileum and colon. **Plus30**, which showed a more promising biological of activity also for its oleuropein content, was characterized by an interesting action against some microorganisms. The results highlighted the ability of **Plus30** to modulate spontaneous and induced contractility, to exert a good antioxidant action, and to significantly act on various microorganisms. These effects are synergistic in the presence of antibiotics. In conclusion, we can confirm that **Plus30** could be a great candidate as a food supplement in patients with IBS.

## 1. Introduction

Irritable bowel syndrome (IBS) is a chronic disease characterized by intermittent abdominal discomfort with diarrhea and constipation in patients without any abnormalities of the digestive tract. This disease is characterized by frequent relapses, which have a negative impact on patients’ quality of life. In the last ten years, there has been an increase of IBS incidence to almost 20% in Europe and America and to 10% in China. Increasingly, 50-year-old women are affected by this disease [1,2]. Taking into account the Rome IV criteria, IBS is characterized by recurrent abdominal pain associated with the following requirements: defecation, change in frequency of stool, and change in form of stool. In addition, these criteria allow for the classification of patients with IBS according to their predominant bowel habit into four categories: IBS with constipation (IBS-C), IBS with diarrhea (IBS-D), mixed type (IBS-M), and unclassified. (IBS-U) [3]. To date, physiological mechanisms linked to IBS are unclear. However, different studies show that altered intestinal microbiota, dysfunctional gastrointestinal motility, stress-induced inflammation, visceral hypersensitivity, brain-gut neuronal axis defects, and psychological factors can occur in IBS patients [1,2]. Furthermore, some evidence suggests that dysbiosis contributes to the onset and symptomatology of IBS [4]. Subjects affected by IBS show an opposite trend in bacteria level; in fact, there is a decrease in Bifidobacterium and Lactobacillus strains and an increase in Firmicutes-to-Bacteroidetes ratio [5]. Moreover, microbiota and its metabolites affect gastrointestinal (GI) motility, altering several pathways involving enteric neurons, glia, or enteric muscularis macrophages. This change in GI motility is one of the marks of IBS [6].

In recent years, the United States Food and Drug Administration (FDA) has approved the use of aminosalicylates, corticosteroids, immunosuppressants, antibiotics motility regulators, and biologic agents to treat IBS patients. On the other hand, the long-term intake of these drugs provokes severe side effects [7]. Due to this, it is imperative to study a new therapeutical approach that includes fecal microbiota transfer and holistic and integrative medicine approaches [3]. Moreover, other strategies, such as diet-related and physical activity-related lifestyle changes, have a positive impact on IBS treatment. [8] The scientific literature shows that diets rich in plant-derived products with a high content of bioactive compounds, mono/polyunsaturated fatty acids, and polyphenols decrease the incidence of metabolic syndrome. In particular, polyphenols are widely available in different diets, and they could act as a good method by which to treat a large amount of diseases [7,8]. *Olea europea* L. polyphenols, for example, have anti-inflammatory and anti-oxidative mechanisms [9] that could be used as adjuvant therapy in IBS [10]. Polyphenols stimulate the microbiota composition playing as prebiotics, in particular, inhibiting pathogenic bacteria growth, such as *E. coli*, and stimulating the probiotic bacteria, such as *Bifidobacterium*. They establish a microbial equilibrium because they reach the colon without being absorbed in the upper gastrointestinal region [11].

The olive tree (*Olea europaea* L.) has been identified as source of several botanical drugs in traditional medicine; besides the fruit (olive), the leaves and the by-products of the milling of the olives also contain, specifically, phenolic derivates, such as phenolic acids, phenolic alcohols (hydroxytyrosol), flavonoids, and secoiridoids (oleuropein). These compounds are involved in antihypertensive [12], antiatherogenic, anti-inflammatory, hypoglycemic, and hypocholesterolemic activities. A deep investigation related to off-target effects on antihypertensive network target shows that olive leaf extract induces interesting spasmolytic activity on ileum and proximal colon [13]. A recent study demonstrated the intestinal anti-inflammatory properties of olive leaf extract experimental models of mouse colitis, applying this waste for a potential application in IBS syndrome [14]. Thanks to these phenolic compounds, including oleuropein, important mechanisms are developed, such as the ability to restore the integrity of the intestinal barrier and the reduction of the expression and/or production of proinflammatory cytokines induced by inflammatory stimuli [7]. The extract obtained from the leaves also combines an interesting effect on the modulation of both spontaneous and induced intestinal contractility, an important target in the control of this intestinal pathology [15]. The secondary metabolites are present also in waste that comes from the production process of olive oil, such as wastewater olive oil and pomace. Indeed, in many cases, various biological properties are associated with the waste resulting from food processing [16]. In this work, we focused our attention on the olive-mill wastewater (OMW), which for ages, has been considered hazardous waste; indeed, it is a potential low-cost material that contains bioactive compounds, particularly phenolics, tocopherols, and carotenoids, that can be used as a natural antioxidant for cosmetic, food, and pharmaceutical industries [17,18]. Some studies have already evaluated the potential biological activity of OMW, and it has been revealed that it has antioxidant and anti-inflammation effects [19], anti-angiogenic and chemopreventive effects [20], neuroprotective activity [21], and cardio-vascular activity [22].

Taking into account the aforementioned benefits, it is necessary to estimate the potential activities of OMW in intestinal inflammatory diseases. In particular, we explored the possible application of some products obtained from wastewater olive oil (**MOMAST^®^**) as therapeutic adjuvant of patients with this disease.

## 2. Materials and Methods

### 2.1. Wastewater Plant Material

A rich, polyphenolic liquid complex, which is derived from the mechanical filtration process of wastewater olive oil, is produced from Apulian olive (*Olea europaea* L., mainly Cultivar Coratina). All **MOMAST^®^** (**Plus30**, **HY100**, and **PW25**) were provided by Bioenutra SRL (Ginosa, Taranto, Italy), the partner of the testing. **Plus30** and **HY100** are liquid complexes that differ in production process [23,24], while the **PW25** is a solid complex that comes from **Plus30**.

#### 2.1.1. Chemical Analysis

HPLC-DAD analysis. For all **MOMAST^®^** (**Plus30**, **HY100**, and **PW25**), the analysis of total phenolic compounds was performed according to the IOC official method [25]. HPLC-DAD analysis was carried out with an HPLC 1200 (Agilent Technologies, Palo Alto, CA, USA) equipped with a degasser, quaternary pump solvent delivery, thermo-stated column compartment, and a diode array detector. Chromatographic separation was performed by using SphereClone ODS (2), 5 μm, 250 × 4.6 mm; the elution solvents were acidified H_2_O by phosphoric acid (pH 3.2), CH_3_CN, and MeOH. The flow rate was 1 mL min^−1^, with an injection volume of 20 µL; the applied gradient was in accordance with the IOC method. The area values were registered at 280 nm with syringic acid as the internal standard. The results were expressed as mg of tyrosol per g of sample.

#### 2.1.2. Determination of the Total Phenolic Content (TPC)

The total amount of polyphenols in the **MOMAST^®^** was determined according to the slightly modified procedure of Blainski [26]. Briefly, 50 µL of each sample was mixed with 50 µL of Folin–Ciocâlteu reagent, 50 µL of MeOH, and 250 µL of water. After 5 min, 200 µL of sodium carbonate (20%) and 400 µL of water were added. The final mix was incubated at 30 °C for 90 min. The absorbance of each sample was measured at 700 nm (Perkin Elmer, Lambda Bio 20, Boston, MA, USA). Gallic acid, in different concentrations (0.025–0.200 mg/mL), was applied as the reference standard; a blank was prepared by using MeOH instead of the samples and the standard. The TPC value was expressed as milligrams of a gallic acid equivalent (GAE) per g of **MOMAST^®^** (mg GAE/g).

Chemicals and Reagents. MeOH, Folin–Ciocâlteu reagent, gallic acid, ascorbic acid, sodium carbonate, 2,2-diphenyl-1-picrylhydrazyl (DPPH), 2,2′-azobis(2-amidinopropane) dihydrochloride (AAPH), 2,4,6-tris(2-pyridyl)-s-triazine 98%, and 6-hydroxy-2,5,7,8-tetramethyl chromane 2-carboxylic acid (Trolox) were purchased from Sigma-Aldrich (Milan, Italy). Formic acid, acetonitrile, and HPLC-grade water used for HPLC analyses were purchased from Sigma-Aldrich (Milano, Italy). Norfloxacin and rifaximin were purchased by Sigma-Aldrich SRL (Milan, Italy). The culture media used are Mueller Hinton Broth (Oxoid, Rodano, Milano, Italy) and Mueller Hinton Agar (Oxoid, Rodano, Milano, Italy).

### 2.2. Antioxidant Activity

DPPH assay. The DPPH radical scavenging assay was determined using the method reported in Clodoveo et al. [27], with some modifications. Briefly, 650 μL of each sample was mixed with 350 μL of DPPH in methanol to obtain a final concentration of 100 μM. The mixture was then vortexed and incubated at 30 °C for 30 min in the dark. The reduction of DPPH was measured at 517 nm using a UV-vis spectrophotometer (Lambda Bio 20, Perkin Elmer, Milano, Italy) Gallic acid, in different concentrations (0–4 μg/mL), was applied as the reference standard; a blank was prepared by using MeOH instead of the samples and the standard. The results of the DPPH assay were expressed as SC_50_ values (μg/mL necessary for 50% reduction of the DPPH radical) ± SD, which were derived from the curve to different concentrations of a sample.

### 2.3. Ex Vivo Contractility Studies

#### 2.3.1. Animals

Guinea pigs of either sex (200–400 g) obtained from Charles River (Calco, Como, Italy) were used. The animals were housed according to the ECC Council Directive regarding the protection of animals used for experimental and other scientific purposes (Directive 2010/63/EU of the European Parliament and of the Council) and the WMA Statement on Animal Use in Biomedical Research. All procedures followed the guidelines of animal care and use committee of the University of Bologna (Bologna, Italy). The ethical committee authorization was reported and numbered as “Protocol PR 21.79.14” by the Comitato Etico Scientifico for Animal Research Protocols according to D.L. vo 116/92.

Guinea pig ileum. As previously described [28], the terminal portion of the ileum (3–4 cm near the ileo-caecal junction) was cleaned, and segments 2–3 cm long of ileum were set up under 1 g tension at 37 °C in organ baths containing Tyrode solution of the following composition (mM): NaCl, 118; KCl, 4.75; CaCl_2_, 2.54; MgSO_4_·7H_2_O, 1.20; KH_2_PO_4_·2H_2_O, 1.19; NaHCO_3_ 25; and glucose 11. The two segments obtained (2–3 cm) were set up under 1 g tension in the longitudinal direction along the intestinal wall. Tissues were allowed to equilibrate for at least 30 min, during which time the bathing solution was changed every 10 min.

Guinea pig proximal colon. Starting approximately 1 cm distal from the caecocolonic junction, two segments of about 1 cm of the guinea pig proximal colon was cut. The proximal colon was cleaned by rinsing with De Jalon solution of the following composition (mM): NaCl, 155; KCl, 5.6; CaCl_2_, 0.5; NaHCO_3_, 6.0; and glucose, 2.8; and the mesenteric tissue was removed. The two segments were suspended in organ baths containing gassed, warm De Jalon solution under a load of 1 g maintained at 37 °C. Tension changes in longitudinal muscle length were recorded. Tissue were allowed to equilibrate for at least 30 min, during which time the bathing solution was changed every 10 min [29].

#### 2.3.2. Spontaneous Contractility

Ileum and proximal colon tracing graphs were recorded: after equilibration, cumulative concentration-response curves of extracts were constructed using Papaverine as positive control.

At the end of each single concentration, a 5 min stationary period was selected, and for each interval, the following parameters were evaluated: mean contraction amplitude (MCA), calculated as the mean force value (g); the force contractions standard deviations, considered as an index of the spontaneous contraction variability (SCV); and basal spontaneous motor activity (BSMC), calculated as the percentage (%) variation of each mean force value (g) with respect the control. The absolute powers of the following frequency bands of interest—low (0.0, 0.2) Hz (LF), medium (0.2, 0.6) Hz (MF), and high (0.6, 1.0) Hz (HF) [5]—were then calculated. The PSD percentage (%) variations for each band of interest with respect to control were estimated [28].

#### 2.3.3. Induced Contractility

Calcium channel-blocking activity. The spasmolytic activity was studied using high K^+^ concentration as previously described [28]. Briefly, after the equilibration period, guinea pig aortic strips were contracted by washing in PSS containing 80 mM KCl (equimolar substitution of K^+^ for Na^+^). When the contraction reached a plateau (about 45 min or 15 min, respectively), different concentrations of the **MOMAST^®^** (0.01–10.0 mg/mL) or Papaverine (0.1–100 µM) were added cumulatively to the bath, allowing for any relaxation to obtain an equilibrated level of force. Tension changes in smooth muscle relaxation by acting on L-type calcium channels were recorded isometrically.

H_1_-receptor antagonism activity. Using ileum tissue, we tested the activity of **MOMASTs** against H_1_-receptor. Concentration-response curves were constructed by cumulative addition of the agonist (Histamine). The concentration of agonist in the organ bath was added only after the response to the previous addition had attained a maximal level and remained steady. Contractions were recorded by means of displacement transducer (FT. 03, Grass Instruments, Quincy, MA, USA) using Power Lab software (ADInstruments Pty Ltd., Castle Hill, Australia). In all cases, parallel experiments in which tissues did not receive any antagonist were run in order to check any variation in sensitivity. Concentration-response curves to agonist were obtained at 30 min intervals, the first one being discarded and the second one taken as control. Following incubation with the antagonist (**MOMASTs**), a new concentration-response curve to agonist was obtained. Papaverine has taken as reference compound. Tension changes were recorded isotonically.

#### 2.3.4. Statistical Analysis

Comparisons between spontaneous contraction amplitudes at different concentrations were performed by one-way ANOVA with Bonferroni’s correction and after the assessment of the data normality through a Kolmogorov–Smirnov Test. Differences with *p* < 0.05 were considered statistically significant. All data of induced contractility are presented as mean ± SD or 95% confidence limits. In all comparisons, a *p*-value less than 0.05 was considered significant.

For the inhibition of contraction induced by K^+^ (80 mM), the potency was calculated from concentration-response curves. Data were analyzed using Student’s *t*-test and are presented as mean ± SD [30]. Since the drugs were added in cumulative manner, the difference between the control and the experimental values at each concentration were tested for a *p*-value < 0.05. The potency of drugs defined as IC_50_ was evaluated from concentration-response curves (Probit analysis using Litchfield and Wilcoxon [30] or GraphPad Prism^®^ software [31,32]) in the appropriate pharmacological preparations. 

For H_1_-antagonist activity, three different **MOMAST^®^** or Papaverine concentrations were always used, and each of them was tested at least four times in at least three different guinea pig ileum preparations. Antagonist activity of each **MOMAST^®^** or Papaverine toward H_1_-receptors was estimated by determining the concentration of each extract that inhibited 50% of the maximum response evoked by histamine and reported as IC_50_ [31]. The potency of **MOMASTs** or Papaverine, reported as IC_50_, was calculated from concentration-response curves obtained by plotting the concentration of each extract vs. the percent of relaxant activity [30].

Statistical significance was assessed by using ANOVA followed by Dunnett post hoc test [31,32].

### 2.4. Antimicrobial Activity

#### 2.4.1. Microorganisms

The antibacterial activity was tested against many bacterial strains, taking into consideration both gram-positive and gram-negative bacteria, and including several strains belonging to the American Type Culture Collection (ATCC, Rockville, MD, USA) or derived from clinical isolation. Strains from the ATCC were *Bacillus cereus* ATCC 10876, *Staphylococcus aureus* ATCC 6538P, *Staphylococcus aureus* ATCC 29213, *Staphylococcus aureus* ATCC 43300 (MRSA), *Staphylococcus aureus* ATCC 25923, *Enterococcus faecalis* ATCC 29212, *Escherichia coli* ATCC 25922, *Escherichia coli* ATCC 35218, *Klebsiella pneumoniae* ATCC 13883, *Acinetobacter baumanni* ATCC 19606, and *Pseudomonas aeruginosa* ATCC 27853. 

Clinical isolates. All the isolates were from patients admitted to the intensive care unit of the Department of Biomedical Science and Human Oncology, University of Bari, Italy. The isolation and identification procedures were conducted at the Hygiene Section of the Department. Using conventional physiological and morphological methods (API systems), the strains were identified as *Staphylococcus aureus* 23, *Staphylococcus aureus* 24, *Staphylococcus aureus* DEL, *Staphylococcus aureus* IAC, *Staphylococcus aureus* TER, *Enterococcus faecalis* 1011, *Enterococcus faecalis* 2011, *Enterococcus faecalis* BS, *Escherichia coli* ESBL BS, *Citrobacter freundii* IG, and *Proteus mirabilis* IG. Multi-drug-resistant (MDR) clinical isolates were *Acinetobacter baumannii* BS and *Klebsiella pneumoniae* BS. The clinical strains come from different patients with disparate pathologies. 

All strains were grown and maintained on Mueller Hinton agar (Oxoid, Italy) at 37 °C.

Preparation of the stock solution. **MOMAST^®^** preparation: 1 mL stock solution was added to 4mL MHB, obtaining the concentration solution 260 mg/mL. Rifaximin preparation: solubilization of 6mg of Rifaximin in 2 mL of DMSO and dilution 1:30 obtaining Rifaximin 100 μg/mL.

#### 2.4.2. Agar Disk-Diffusion Method

This procedure, recommended by Clinical and Laboratory Standards Institute (CLSI) [33], means that agar plates are inoculated with a standardized inoculum of the test microorganism. Then, two filter paper discs, one containing 20 µL of **MOMAST^®^** concentration solution (260 mg/mL) and other one containing 20 µL of Norfloxacin, are placed on the agar surface. The Petri dishes are incubated under suitable conditions. In general, antimicrobial agent diffuses into the agar and inhibits growth of the microorganism and then the diameters of inhibition growth zones are measured.

Moreover, this method is not used to determine the Minimum Inhibitory Concentration (MIC) since it does not allow to quantify the antimicrobial agent diffused into the agar medium.

#### 2.4.3. Determination of the Minimum Inhibitory Concentration (MIC) 

The Minimum Inhibitory Concentration (MIC) was determined by the broth microdilution method as recommended by CLSI [33]. A bacterial sample was taken from previously cultured Petri dishes and diluted in test tubes containing 3 mL of MHB (Mueller Hinton Broth, Sigma-Aldrich, St. Louis, MO, USA). Then, serial dilutions were performed with 100 μL of the test solutions to obtain concentrations range from 130 to 0.1 mg/mL for **MOMAST^®^** and from 50 to 0.05μg/mL for Rifaximin until the penultimate well and the last one were able to control microbial growth [34]. The plates were then incubated at 35 °C for 24 h. The MIC was defined as the lowest concentration at which no microbial growth was observed. MIC determinations were realized in triplicate in three independent assays.

#### 2.4.4. FICI Determination

MIC data of the Rifaximin and **MOMAST^®^** were converted into Fractional Inhibitory Concentration (FIC), determined by using the formula FIC = (MICA combination/MICA alone). MIC values for the **MOMAST^®^**—Rifaximin were defined as the lowest concentration at which no visible growth of the microbial strains could be detected compared to their growth in the control well as described in Eucast document [35].

#### 2.4.5. Microdiluition Checkerboard Method

In the combination assays, the checkerboard procedure described by White et al. [36] was followed to evaluate the synergistic action of the **MOMAST^®^** and Rifaximin. This method was used to mix each concentration of Rifaximin with the appropriate concentrations of **Plus30** in order to obtain a series of combinations between the two substances. In our experimental protocol, the substance combinations were analyzed by calculating the FIC index (FICI) as follows: FIC of the **MOMAST^®^**, plus FIC of the Rifaximin. Overall, the FICI value was interpreted as: a synergistic effect when ≤0.5; an additive or indifferent effect when >0.5 and <1; an antagonistic effect when >1 [36].

The concentrations made for the **MOMAST^®^** are constituted for 40%, 20%, 10%, 5%, 2.5%, 1.25%, 0.6%, and 0.3% of the MIC value and 25%, 12.5%, 6.25%, 3.12%, 1.56%, 0.8%, 0.4%, and 0.2% of the MIC value for the antibiotic.

## 3. Results

### 3.1. Chemical Composition of MOMASTs

Total Polyphenol Content (TPC) of **MOMAST^®^** (**Plus30**, **HY100**, and **PW25**), determined by Folin–Ciocâlteu assay, was carried out considering the absorbance of each sample at 700nm and the gallic acid standard solution calibration curve: y = 5.5923x − 0.0005. Table 1 shows that the **MOMAST^®^ Plus30** (85.61 ± 2.33) has the highest absolute value of TPC, followed by **MOMAST^®^ HY100** (53.4 ± 1.21) and **MOMAST^®^ PW25** (29.31 ± 0.62).

The chemical composition of all **MOMAST^®^** (**Plus30**, **HY100**, and **PW25**), obtained by HPLC-DAD analysis, is reported in Table 1. The most abundant phenols in each sample are hydroxytyrosol and tyrosol; only in **MOMAST^®^ Plus30** did we identify oleuropein, a compound known to have a great deal of biological activity [37,38,39,40].

### 3.2. Antioxidant Activity

The antioxidant activities of **MOMAST^®^** (**Plus30**, **HY100**, and **PW25**) were evaluated using the DPPH assay.

The DPPH radical scavenging activity of **MOMAST^®^** was dose dependent as shown in Figure 1, and the results, expressed as SC_50_, are reported in Table 2 with those of gallic acid (GA) taken as reference compound. **HY100** (4.65 μg/mL) showed the highest antioxidant activity, followed by **Plus30** (5.44 μg/mL), while **PW25** (6.29 μg/mL) showed the lower activity. Interestingly, gallic acid, taken as a positive control, is only about two times more potent than **MOMAST^®^** (**HY100**: 1.6, **Plus30**: 1.9, and **PW25**: 2.2 times, respectively) as radical scavenging. These data are particularly important, as gallic acid is a single molecule, while the components of **MOMAST^®^** are mixtures of compounds as proof of their strong antioxidant action.

### 3.3. Effect on Ileum and Colon Ex Vivo Contractility

#### 3.3.1. Induced

All **MOMAST^®^** and Papaverine, taken as a positive control, were studied for their spasmolytic effects on the contraction induced by 80 mM potassium. In these experimental conditions, it is possible to verify the ability of **MOMAST^®^** to interfere with the movements of the calcium through the dedicated channels [12]. As can be seen from the data collected in Table 3, **Plus30** is the most interesting extract, as it is active both on the ileum and on the colon. The spasmolytic action is significantly different from that of **HY100** both on the ileum and on the colon. As for **PW25**, it has an action on ileum with a potency not significantly different from **Plus30** but has no significant effects on the colon, and the intrinsic activity is, in fact, 15% at the maximum concentration studied.

Figure 2 highlights how all **MOMAST^®^** compounds respond in a concentration-dependent manner as the reference compound.

All extracts have been studied for their ability to block the H_1_-histaminergic receptor. The experiments were conducted on isolated guinea pig ileum. All extracts were studied at 1mg/mL. The results are shown in Figure 3 together with those obtained with Papaverine (0.01 mg/mL) taken as a positive control. As can be seen, all **MOMAST**^®^ compounds have a weak, non-competitive antagonist action against the H_1_-histaminergic receptor, with effects of around 30%; in particular, we recorded these values: **Plus30** (24.0 ± 1.5), **HY100** (32.0 ± 1.4) and **PW25** (39.0 ± 2.9). Papaverine inhibits the contraction induced by histamine by 81.0 ± 2.7 at 0.01 µM. Although **MOMAST**^®^ compounds are less potent than Papaverine, their effects are reversible with 30 min of washing, similarly as for the reference compound (data not shown).

#### 3.3.2. Spontaneously

The **MOMAST**^®^ compounds were studied for their effects on ileum and colon spontaneous contractility using Papaverine as a positive control [41].

*Ileum*. All **MOMAST**^®^ compounds raised the tone (Figure 4). In particular, **Plus30** increased up to a concentration of 1 mg/mL and then remained almost constant. For **HY100** and **PW25**, the tone increased significantly for almost all concentrations. Papaverine did not bring significant changes in the tone of the ileum smooth muscle (Figure 5). As for the waves of peristalsis (PSD), **MOMAST**^®^ compounds behave in a different way. **Plus30** has a biphasic behavior: up to 1 mg/mL the frequencies increase, while at higher concentrations, a decrease in tone is observed. Both **HY100** and **PW25** cause an increase in the mid frequencies.

Colon. All **MOMAST^®^** compounds cause a decrease in tone up to a concentration of 1 mg/mL (Figure 6). Papaverine also significantly reduces tone in a concentration-dependent manner (Figure 7). All **MOMAST^®^** compounds in general reduce PSD, with important effects at concentrations higher than 0.5 mg/mL. Papaverine reduces all frequencies until 1 mg/mL.

The results related to ileum and colon spontaneous contractility are summarized in Table 4.

### 3.4. Effect of **MOMAST^®^ Plus30** against Bacteria

Given the behavior of the **MOMAST^®^**, given their composition, and taking into account that the **PW25** does not exert significant actions on the colon, the antimicrobial activity was first assessed only on **Plus 30**. The bacteria were analyzed for susceptibility to **Plus30** by using the agar disk diffusion method, which was performed according to procedure M7 A10 from the CLSI [33]. Through agar diffusion, **Plus30** produced inhibition zones against all strains (Table 5). The most sensitive strains are *Staphylococcus aureus* both ATCC and clinical isolation. 

#### 3.4.1. Determination of the Minimum Inhibitory Concentration (MIC)

The results of a minimum inhibitory concentration of **Plus30** and control (Rifaximin) are shown in Table 6. ATCC strains respond better than strains from clinical isolation, and rifaximin values are in line with international protocols [33].

#### 3.4.2. FICI Determination

Best results are achieved in synergy between **Plus30** and Rifaximin (Table 7). Indeed, the data obtained clearly show a significant reduction in antibiotic concentration when used in combination with **Plus30**. The MIC_50_ values of Rifaximin in association with **Plus30** are reduced, for most of the cases, by 500 times. These results underlined the large reduction of the quantities of Rifaximin and **Plus30** used to achieve the association with respect to the quantity of the substances used alone to inhibit the strains.

## 4. Discussion

Chronic inflammatory bowel syndromes (IBS) are characterized by inflammatory lesions that mainly affect the ileum and colon. The therapy involves a modification of the diet combined with anti-inflammatory pharmacological treatments, such as aminosalicylates and cortisone. These drugs can be associated with contractility correctors and antibiotics [1].

A paper by Recinella et al. [19] highlighted that **MOMAST^®^** exerts a protective effect against induced inflammatory insults. To these effects, **MOMAST^®^** combines an interesting preventive action against induced oxidative stress, in particular at the level of the colon. As reported, all **MOMAST^®^** have an interesting radical-scavenging capacity (Figure 1). This action is important in diseases such as IBS, where the presence of oxidative stress is known and probably due to numerous factors [42]. In this regard, the literature reports the use of medicinal plants with antioxidant action in patients with IBS [43]. In addition to these, functional foods, such as EVOO, are also recommended because they activate antioxidant enzymes through the presence of polyphenols [44] represented in large quantities in **MOMAST^®^**. Hence, the possibility of applying **MOMAST^®^** in IBS can be considered. Among all **MOMAST**^®^ compounds, the presence in **MOMAST^®^ Plus30** of interesting quantities of compounds with a phenolic structure and of oleuropein led us to explore their actions on other targets connected to IBS. This pathology is also characterized by motility disorders, and oleuropein has an action on calcium channels both as an isolated molecule and if used in the form of phytocomplex [12].

This effect on calcium channels is manifested by spasmolytic action on the ileum and colon.

In particular, **Plus30** is active in both districts in analogy with Papaverine. These effects also affect spontaneous contractility: **Plus30** in fact slightly reduces the tone with possible consequent effects on peristalsis and could regulate the progress of the food bolus with possible consequences on promoting the absorption of nutrients.

**Plus30** combines this effect with an increase in the low frequencies. The effects of other **MOMAST^®^** samples show that, for spontaneous contractility, **PW25** leads to an increase in tone accompanied by an increase in contractility at low frequencies. The increase in low frequencies compensates for the increase in tone and probably does not significantly change the progression of the food bolus in the ileum. For **HY100**, the increase in tone together with the increase in medium frequencies could be induce an increase in the speed of transit and spasms related to pain. Papaverine on the ileum probably reduces absorption with an increase in spasms; in fact, it increases the low and medium frequencies without a significant increase in tone. The effects of all **MOMAST^®^** and Papaverine on the colon are less marked than those seen on the ileum. **HY100** reduces tone and frequencies: these combined effects probably do not cause changes in the progression of the food bolus while reducing spasms at the same time. The same activity profile is observed for both **PW25** and **Plus30** but for the latter at concentrations greater than or equal to 0.5 mg/mL. Papaverine has a similar profile but causes an increase in low frequencies to 5 μM; this could cause an increase in transit speed. In order to also investigate the **MOMAST^®^** effect on histamine that, as reported, can play a role in gut disorders, the H_1_-histaminergic receptor affinity was studied. 

Histamine is a biogenic amine produced physiologically also by some bacteria. Excess histamine causes mainly allergic systemic effects that involve various organs. In the intestine, among many other effects, it causes abdominal pain, flatulence, diarrhea, and inflammation. The inflammatory action following an allergic reaction is also supported by the release of histamine from the mast cells [45]. Mast cells are in fact important mediators for allergic reactions and also intervene in inflammatory processes. The release of histamine is a direct consequence of their activation [46]. Among all the inflammatory mediators, histamine exerts various actions, including the vasodilation that characterizes the acute phase [47]. Recent studies show an increase of these cells in patients with IBS also confirmed in animal models [48]. For all these reasons, histamine plays a key role in functional gut disorder [49]. H_1_-histaminergic receptor blockers are used to reduce visceral hypersensitivity and symptoms in IBS [50]. With these premises, it becomes interesting to evaluate the ability of **MOMAST^®^** to block the H_1_-histaminergic receptors present in the intestine. All **MOMAST^®^s** are able to exert a reversible non-competitive antagonist effect on the H_1_-histaminergic receptor (Figure 3).

The potency is significantly different from that of Papaverine taken as a reference compound, but it should not be forgotten that **MOMAST****^®^** is a phytocomplex, while Papaverine is a single molecule. These effects are also important in the presence of histamine-producing bacteria [51] and when we have an overgrowth of intestinal bacteria that can impair digestion. In addition, the use of proteins with significant amounts of histidine leads to the production of histamine, compromising clinical symptomatology.

Certainly, the choice of correct dietary proteins is important to prevent the possible formation of histamine but also prevent its action in the presence of Small Intestinal Bacterial Overgrowth, SIBO [52,53]. In addition, the antagonistic activity against histamine receptors in the gastrointestinal tract also attenuates the indirect effects of a potential intestinal dysbiosis that is frequent in these pathologies [54].

IBS is associated with microbiota dysbiosis, often characterized by intestinal infections supported by pathogenic bacteria. The use of antibiotics, necessary in these cases, aggravates the dysbiosis and, in patients with IBS, worsens the symptoms. It is therefore urgent to identify bioactive compounds that, in synergy with antibiotics, can reduce their concentration in order to also reduce the impact on the intestinal microbiota. To this end, in the study, the antimicrobial action of the **MOMAST^®^ Plus30** was evaluated on human bacterial strains of gram-negative and gram-positive pathogens of international collection and clinical isolates because, among all **MOMAST^®^**, it could be a candidate as adjuvant in the spasmolytic therapy of IBS.

The results obtained confirm a resolutive antibacterial action of **Plus30** (MIC) in the range of 16–32 mg/mL.

None of the bacterial strains tested in both Agar diffusion and microdilution have demonstrated any form of resistance to **Plus30**. The biologically more resistant gram-negative bacterial strains to the action of the substances have, however, found an accentuated sensitivity even if with values of MIC of lesser effectiveness.

The strains that come from clinical isolation demonstrate greater resistance to the product but are, however, more able to resist the action of the product. 

The activity of the product is absolutely encouraged and improved when it is associated with the antibiotic Rifaximin with extremely low FICI values, ranging from 0.01 to 0.05 for all bacterial strains evaluated. 

In terms of percentage, the quantity of product present in the association compared to the MIC values goes from a minimum of 0.3% up to 5.02%. The percentage values of the amounts of antibiotic used compared to an MIC range from 0.2% to 3.12%. 

The reduction of the **Plus30** to the MIC value is highly significant; in fact, it goes from 20 times to 333 times. Equally important is the reduction of the amount of antibiotic, ranging from 32 to 526 times and over for some strains. 

These reduction values take on an important significance in relation to the quantities used and the marked improvement of the inhibitory and bactericidal action.

As for the FICI, the values range from 0.01 to 0.05. Considering that according to Eucast, the synergy value is significant for values below 0.5, the results obtained demonstrate a strong and unequivocal reduction of the relative quantities present in associations.

**Plus 30** showed an important efficacy in the synergistic test towards the strains coming from clinical isolation: the gram-positive and the gram-negatives strains are highly sensitive to the associations of the two products. 

The reductions of the two quantities of products present in the association are comparable to those observed for strains from the ATCC collection. 

It should also be noted the drastic decrease in the amount of antibiotic does not affect the overall bactericidal action. In our experiments, we always recorded bactericidal activity.

The mechanism involved definitely calls into question the ability of polyphenols present in the product to interact with the bacterial cell wall, altering its structure.

The results obtained underline the importance of the **Plus30** plant-based substance, able to inhibit and destroy different bacterial species considered and the strong synergy with the antibiotic rifaximin. These experiments have shown the safe efficacy of association in terms of bactericidal action.

## 5. Conclusions

In conclusion, based on the data presented, we can assume that **MOMAST^®^** compounds are a great candidate to become a food supplement in the treatment of IBS. In particular, **Plus30**, because of its high concentration of polyphenols and oleuropein, showed the most interesting actions on the targets studied relating to the antioxidant action and to both spontaneous and induced contractility. In addition, microbiological test data show that **Plus30** of plant origin develops in vitro a clear antibacterial action that is most enhanced when it is in association with the antibiotic Rifaximin. All our data and in particular those relating to antimicrobial activity are in line with numerous clinical studies aimed at demonstrating the efficacy of complementary therapy in controlling this pathology and related symptoms [55]. Last but not least, studies aimed at verifying any toxicological effects and any off-target actions in food supplement will be necessary.

## Figures and Tables

**Figure 1 nutrients-14-01264-f001:**
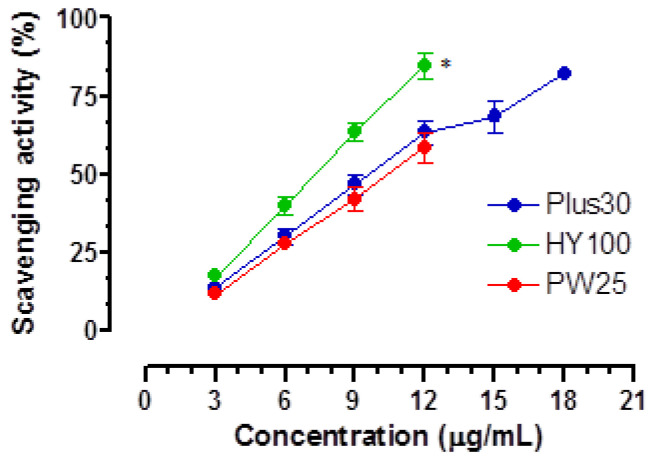
Cumulative dose-response curves of free radical scavenging activity of **MOMAST^®^** (**Plus30**, **HY100**, and **PW25**) measured using DPPH test. Each point is the mean ± SD (*n* = 5–6). * *p* < 0.01 **HY100** vs. **Plus30** and **PW25** (ANOVA followed by Dunnett post hoc test). When error bars are not shown, these are covered by the point.

**Figure 2 nutrients-14-01264-f002:**
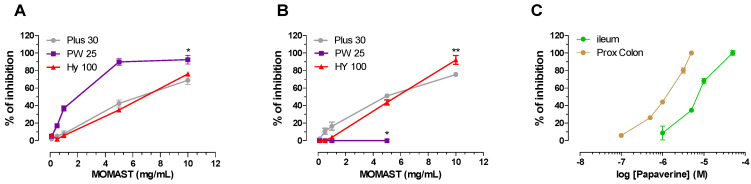
Cumulative concentration-response curves for **MOMAST^®^** (**Plus30**, **HY 100**, and **PW 25**) on K^+^ (80 mM)—induced contraction on isolated guinea pig ileum (**A**) and proximal colon (**B**). Cumulative concentration-response curves curve for Papaverine on ileum and proximal colon (**C**). Each point is the mean ± SD (*n* = 5–6). * *p* < 0.01 **PW25** vs. **Plus30** and **HY100**. ** *p* < 0.05 **PW25** vs. **Plus30** and **HY100** (ANOVA followed by Dunnett post hoc test). Where error bars are not shown, these are covered by the point itself.

**Figure 3 nutrients-14-01264-f003:**
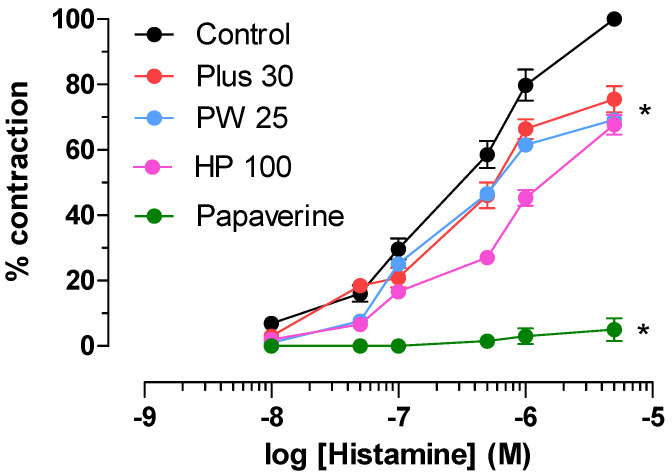
Effect of **MOMAST^®^** (**Plus30**, **HY100**, and **PW25**) and Papaverine on histamine-induced contraction on isolated guinea pig ileum. Cumulative dose-response curves were obtained before and after exposure to **MOMAST**^®^ compounds (1 mg/mL) and Papaverine (0.01 µM) for 30 min. Activity on H_1_-receptor was expressed as % of inhibition at indicated concentration of histamine-induced contraction (Probit analysis by Litchfield and Wilcoxon, with *n* = 4–6) [30]. Each point is the mean ± SD (*n* = 5–6). * *p* < 0.05 **PW25**, **Plus30**, **HY100**, and Papaverine vs. control. (ANOVA followed by Dunnett post hoc test). When error bars are not shown, these are covered by the point.

**Figure 4 nutrients-14-01264-f004:**
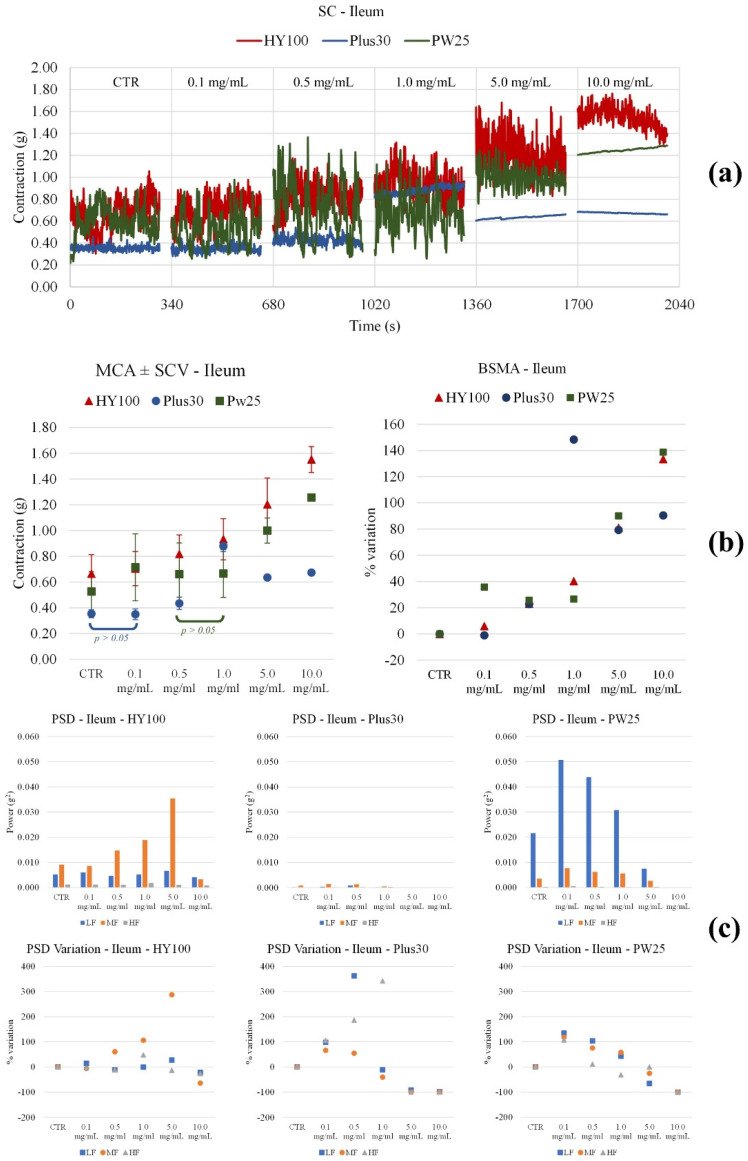
Experimental original recording of the concentration-response curve of **MOMAST^®^** (**Plus30**, **HY100**, and **PW25**) on spontaneous ileum basal contractility. (**a**) Spontaneous contraction (SC) signals for each concentration; (**b**) mean contraction amplitude (MCA) and spontaneous contraction variability (SCV), represented as error bars in the MCA plot and contraction percentage variation with respect to the control (BSMA) for each considered condition; not significant differences (*p* > 0.05) between MCAs at different concentrations are reported in the graph. All the comparisons not reported are to be considered significant (*p* < 0.05); (**c**) absolute powers (PSD) of the different bands of interest (LF: (0.0, 0.2) Hz; MF: (0.2, 0.6) Hz; HF: (0.6, 1.0) Hz) and PSD% variations with respect to the control phase.

**Figure 5 nutrients-14-01264-f005:**
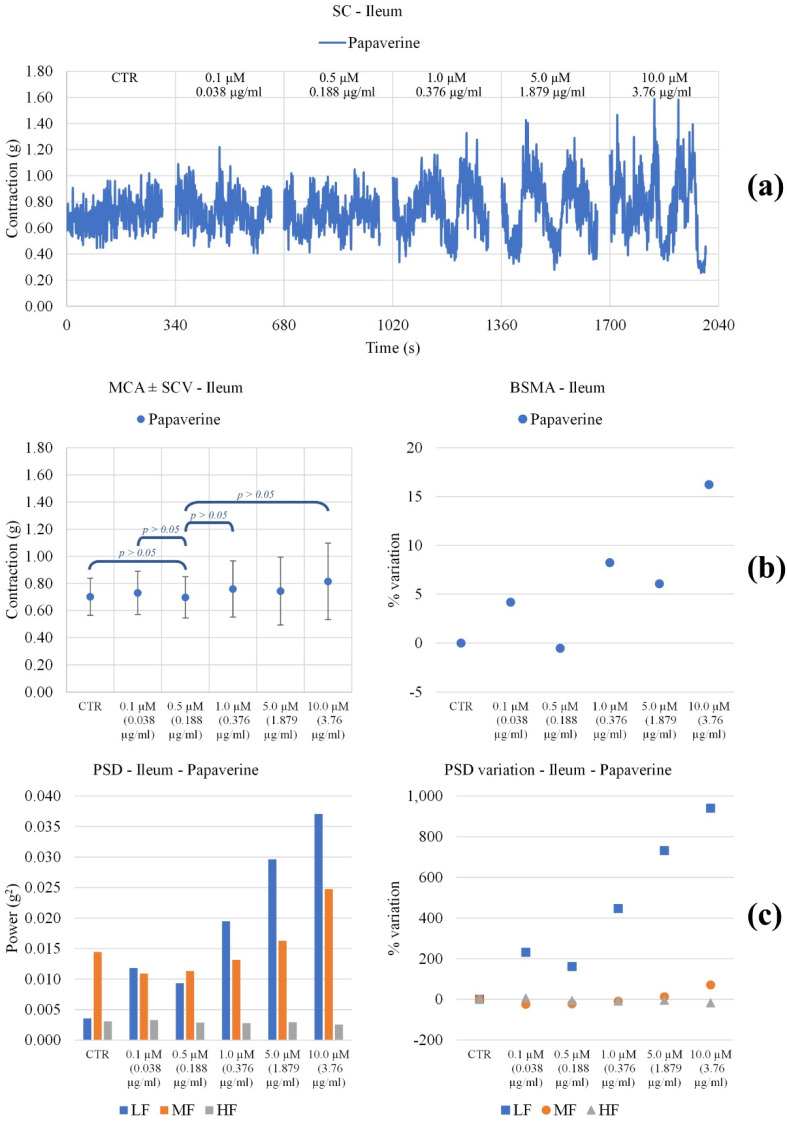
Experimental original recording of the concentration-response curve of **Papaverine** on spontaneous ileum basal contractility. (**a**) Spontaneous contraction (SC) signals for each concentration; (**b**) mean contraction amplitude (MCA) and spontaneous contraction variability (SCV), represented as error bars in the MCA plot and contraction percentage variation with respect to the control (BSMA) for each considered condition; not significant differences (*p* > 0.05) between MCAs at different concentrations are reported in the graph. All the comparisons not reported are to be considered significant (*p* < 0.05); (**c**) absolute powers (PSD) of the different bands of interest (LF: (0.0, 0.2) Hz; MF: (0.2, 0.6) Hz; HF: (0.6, 1.0) Hz) and PSD% variations with respect to the control phase.

**Figure 6 nutrients-14-01264-f006:**
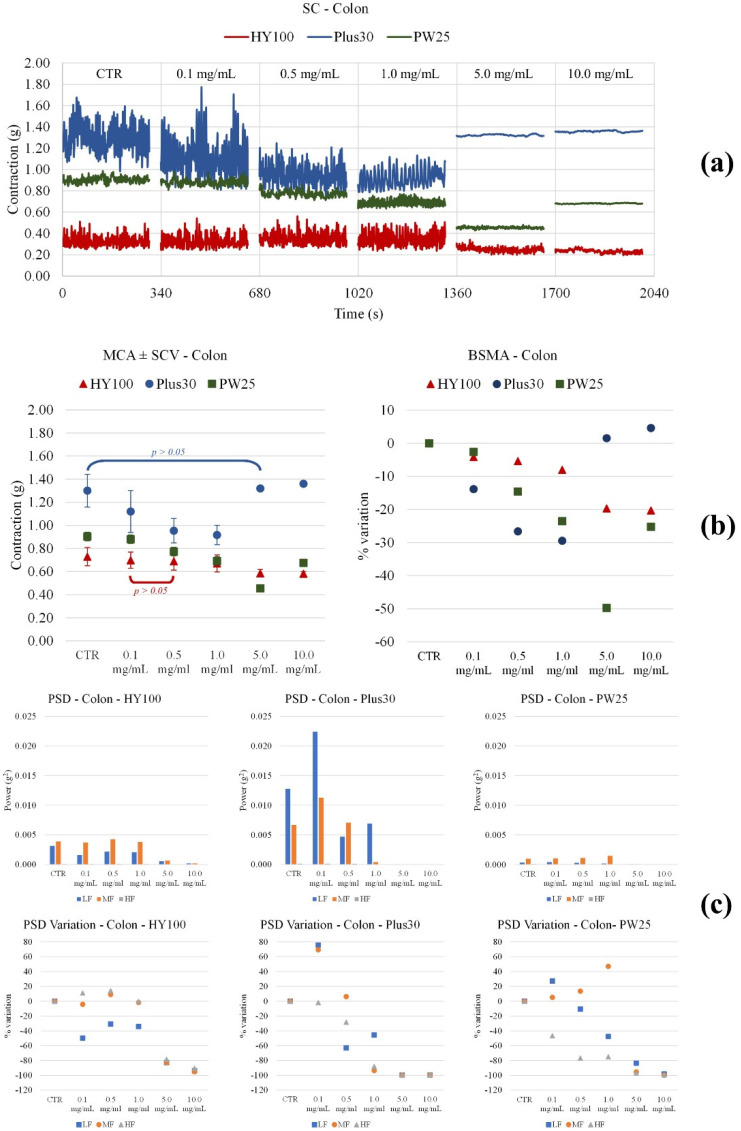
Experimental original recording of the concentration-response curve of **MOMAST^®^** (**Plus30**, **HY100**, and **PW25**), on spontaneous proximal colon basal contractility. (**a**) Spontaneous contraction (SC) signals for each concentration; (**b**) mean contraction amplitude (MCA) and spontaneous contraction variability (SCV), represented as error bars in the MCA plot and contraction percentage variation with respect to the control (BSMA) for each considered condition; not significant differences (*p* > 0.05) between MCAs at different concentrations are reported in the graph. All the comparisons not reported are to be considered significant (*p* < 0.05); (**c**) absolute powers (PSD) of the different bands of interest (LF: (0.0, 0.2) Hz; MF: (0.2, 0.6) Hz; HF: (0.6, 1.0) Hz) and PSD% variations with respect to the control phase.

**Figure 7 nutrients-14-01264-f007:**
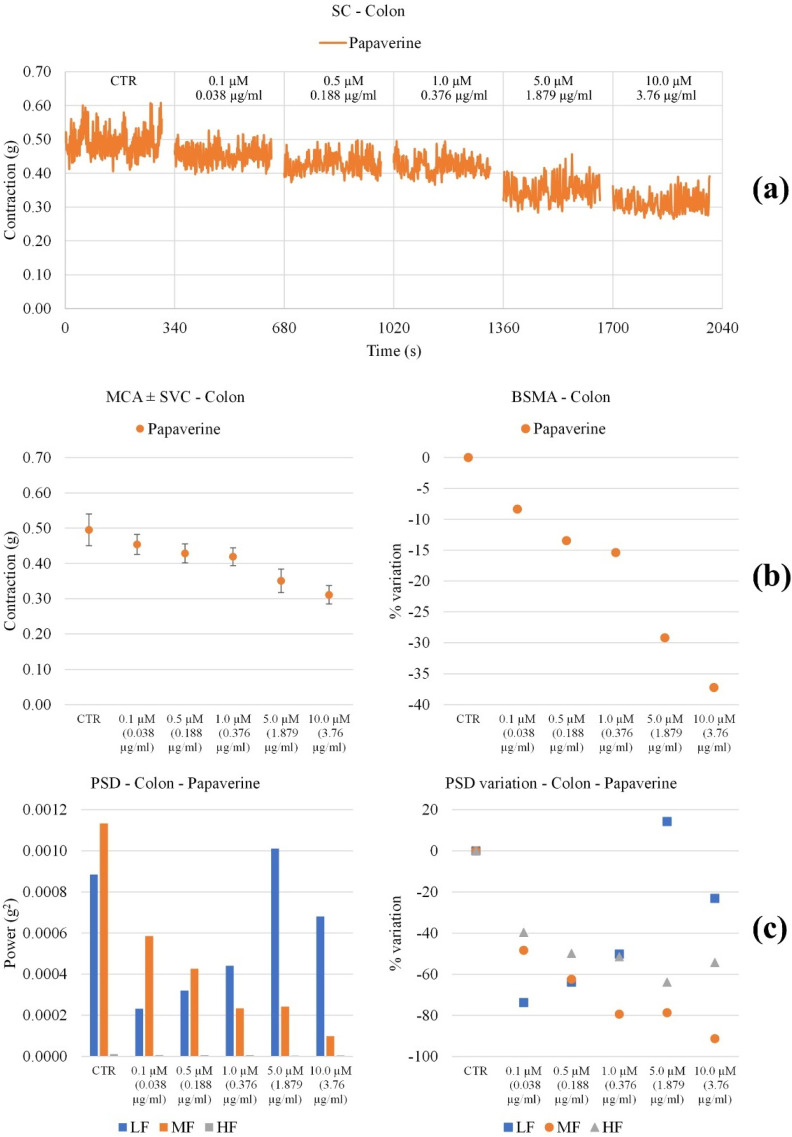
Experimental original recording of the concentration-response curve of **Papaverine** on spontaneous proximal colon basal contractility. (**a**) Spontaneous contraction (SC) signals for each concentration; (**b**) mean contraction amplitude (MCA) and spontaneous contraction variability (SCV), represented as error bars in the MCA plot and contraction percentage variation with respect to the control (BSMA) for each considered condition; not significant differences (*p* > 0.05) between MCAs at different concentrations are reported in the graph. All the comparisons not reported are to be considered significant (*p* < 0.05); (**c**) absolute powers (PSD) of the different bands of interest (LF: (0.0, 0.2) Hz; MF: (0.2, 0.6) Hz; HF: (0.6, 1.0) Hz) and PSD% variations with respect to the control phase.

**Table 1 nutrients-14-01264-t001:** Total phenolic and antioxidant activity of **MOMAST^®^** (**Plus30**, **HY100**, and **PW25**).

MOMAST^®^	TPC ^a^	Total Phenols ^b^	Hydroxytyrosol ^b^	Tyrosol ^b^	Oleuropein ^b^
**Plus30**	85.61 ± 2.33	36.9 ± 0.19	25.88 ± 0.11	5.04 ± 0.02	1.80 ± 0.20
**HY100**	70.54 ± 1.95	53.4 ± 1.21	38.32 ± 0.03	8.40 ± 0.03	-
**PW25**	0.48 ± 0.01	29.31 ± 0.62	21.46 ± 0.03	3.20 ± 0.01	-

^a^ Total Phenolic Content, expressed as mg Gallic Acid Equivalent GAE/g **MOMAST^®^**. ^b^ The amount as expressed as mg Tyr/g **MOMAST^®^**. Values represent means ± SD with *p* < 0.05 (*n* = 3).

**Table 2 nutrients-14-01264-t002:** Antioxidant activity of **MOMASTs**.

		DPPH ^a^
**MOMAST^®^**	**Plus30**	5.44 ± 0.30
	**HY100**	4.65 ± 0.22
	**PW25**	6.29 ± 0.05
**GA ^b^**		2.85 ± 0.01

^a^ SC_50_, radical scavenging activity [concentration expressed in μg/mL necessary for 50% reduction of 2,2-Diphenyl-1-Picrylhydrazyl (DPPH) radical]. Values represent means ± SD with *p* < 0.05. ^b^ Gallic Acid positive control.

**Table 3 nutrients-14-01264-t003:** Activity of **MOMAST^®^** (**Plus30**, **HY100**, and **PW25**) on K^+^ (80 mM) induced contraction.

	Ileum			Colon		
MOMAST^®^	Activity *^a^*	IC_50_ *^b^* (mg/mL)	95% conf lim	Activity *^a^*	IC_50_ *^b^* (mg/mL)	95% conf lim
**Plus30**	81 ± 2.7	2.30	1.10−2.80	75 ± 3.9	2.02	1.93−2.51
**HY100**	76 ± 3.9	4.26	3.10−5.18	92 ± 2.4	3.56	2.78−3.70
**PW25**	92 ± 2.4	1.15	0.96−1.37	15 ± 1.2		
				(*5*)		
**Papaverine**	68 ± 3.3	0.0026	0.0022–0.0027	80 ± 4.1	0.00038	0.00033–0.00046
	(*0.0037* mg/mL)	(mg/mL)	(mg/mL)	(0.011 mg/mL)	(mg/mL)	(mg/mL)
	(*10* μM)	6.86	5.96–6.34	(*8* μM)	1.02	0.89–1.24
		(μM)	(μM)		(μM)	(μM)

^*a*^ Percent inhibition of calcium-induced contraction on K^+^-depolarized (80 mM) guinea pig ileum and proximal colon strips at 10 mg/mL concentration presented as mean ± SD with *p* < 0.05. If the maximum effect is obtained at a different concentration, it is indicated in parentheses. ^*b*^ Calculated from concentration-response curves (Probit analysis by Litchfield and Wilcoxon [30] with *n* = 6−7). When the maximum effect was <50%, the IC_50_ values were not calculated.

**Table 4 nutrients-14-01264-t004:** Summary of the effects of **MOMAST^®^** on ileum and colon spontaneous contractility.

MOMAST^®^	Parameter	Ileum	Colon
**Plus30**	Contraction amplitude	Increases significantly from 0.5 mg/mL	Decreases significantly up to 1.0 mg/mL, then returns to initial values
	LF Band Power	Increases up to 0.5 mg/mL, then decreases	Increases at 0.1mg/mL, then decreases
	MF Band Power	Slightly increases at 0.1mg/mL, then decreases	Increases at 0.1mg/mL, then decreases
	HF Band Power	Increases significantly from 0.5 mg/mL	Decreases from 0.1 mg/mL
**HY100**	Contraction amplitude	Increases significantly with concentration	Slightly decreases up to 5.0 mg/mL
	LF Band Power	No important modifications	Decreases from 5.0 mg/mL
	MF Band Power	Increases up to 5.0 mg/mL	Decreases from 5.0 mg/mL
	HF Band Power	No important modifications	Decreases from 5.0 mg/mL
**PW25**	Contraction amplitude	Increases at 0.1 mg/mL, then decreases	Decreases from 0.1 mg/mL
	LF Band Power	Increases at 0.1 mg/mL, then decreases	Increases up to 1 mg/mL
	MF Band Power	Increases at 0.1 mg/mL, then decreases	Decreases up to 5.0 mg/mL
	HF Band Power	Increases at 0.1 mg/mL, then decreases	Decreases significantly with concentration
**Papaverine**	Contraction amplitude	Not significant modifications	Decreases significantly with concentration
	LF Band Power	Slightly decreases up to 0.5 µM, then increases	Decreases up to 5.0 µM, then increases
	MF Band Power	Increases with concentration	Decreases up to 1.0 µM, then increases
	HF Band Power	No important modifications	Decreases up to 5.0 µM then increases

**Table 5 nutrients-14-01264-t005:** Antimicrobial activity of **MOMAST^®^ Plus30** and Norfloxacin against both strains ATCC and clinical isolation.

Strain	Plus30 ^a^	Norfloxacin ^a^
**ATCC**		
**Gram +**		
*Bacillus cereus* (10876)	2.1	1.9
*Enterococcus faecalis* (29212)	2.0	1.9
*Staphylococcus aureus* (25923)	2.5	2.2
*Staphylococcus aureus* (29213)	2.5	2.1
*Staphylococcus aureus* (43300)	2.5	2.0
*Staphylococcus aureus* (6538P)	2.5	2.3
**Gram −**		
*Acinetobacter baumannii* (19606)	1.6	1.5
*Escherichia coli* (25922)	1.9	3.0
*Escherichia coli* (35218)	1.9	2.8
*Klebsiella pneumoniae* (13883)	1.8	1.6
*Pseudomonas aeruginosa* (27853)	1.8	2.5
**Clinical isolation ^b^**		
**Gram +**		
*Enterococcus faecalis* BS	1.9	2.0
*Enterococcus faecalis* RM 1011	1.8	2.0
*Enterococcus faecalis* RM 2011	1.8	2.0
*Staphylococcus aureus* 23	2.0	2.0
*Staphylococcus aureus* 24	2.0	2.0
*Staphylococcus aureus* DEL	2.0	2.0
*Staphylococcus aureus* IAC	2.0	2.0
*Staphylococcus aureus* TER	2.0	2.0
**Gram −**		
*Acinetobacter baumannii* BSR	1.5	1.5
*Citrobacter freundii* BS	1.9	1.5
*Escherichia coli* ESBL	1.0	2.5
*Klebsiella pneumoniae* BSR	1.2	1.6
*Proteus mirabilis* BS	1.7	1.4

^a^ Inhibition zone diameter (expressed as cm). ^b^ The acronyms refer to the disease of the patients from which the microorganisms were taken.

**Table 6 nutrients-14-01264-t006:** Minimum Inhibitory Concentration (MIC) of **Plus30** and control.

Strain	Plus30 (mg/mL)	Rifaximin (μg/mL)
**ATCC**		
**Gram +**		
*Bacillus cereus* (10876)	16.20	0.05
*Enterococcus faecalis* (29212)	16.20	0.05
*Staphylococcus aureus* (25923)	4.00	0.10
*Staphylococcus aureus* (29213)	4.00	0.05
*Staphylococcus aureus* (43300)	2.00	3.12
*Staphylococcus aureus* (6538P)	2.00	0.05
**Gram** −		
*Acinetobacter baumannii* (19606)	16.20	12.50
*Escherichia coli* (25922)	32.50	6.25
*Escherichia coli* (35218)	32.50	6.25
*Klebsiella pneumoniae* (13883)	4.00	12.50
*Pseudomonas aeruginosa* (27853)	32.50	6.25
**Clinical isolation ^a^**		
**Gram +**		
*Enterococcus faecalis* BS	16.20	0.40
*Enterococcus faecalis* RM 1011	16.20	0.10
*Enterococcus faecalis* RM 2011	16.20	0.10
*Staphylococcus aureus* 23	32.50	0.40
*Staphylococcus aureus* 24	32.50	0.40
*Staphylococcus aureus* DEL	32.50	0.02
*Staphylococcus aureus* IAC	32.50	0.05
*Staphylococcus aureus* TER	32.50	0.05
**Gram −**		
*Acinetobacter baumannii* BSR	65.00	25.00
*Citrobacter freundii* BS	32.50	12.50
*Escherichia coli* ESBL	32.50	6.20
*Klebsiella pneumoniae* BSR	65.00	25.00
*Proteus mirabilis* BS	16.20	6.20

^a^ The acronyms refer to the disease of the patients from which the microorganisms were taken.

**Table 7 nutrients-14-01264-t007:** FICI characterized by the synergy of Rifaximin and **Plus30**.

Strain	Plus30 (mg)	Rfx (μg)	Plus30 %	Rfx %	FICI	Reduction Plus30	Reduction Rfx
**ATCC**							
**Gram +**							
*Bacillus cereus* (10876)	16.25	0.05	5.00	0.20	0.05	20.00	500.00
*Enterococcus faecalis* (2921)	16.25	0.05	1.25	0.20	0.01	80.00	500.00
*Staphylococcus aureus* (25923)	4.00	0.10	0.30	0.80	0.01	333.30	125.00
*Staphylococcus aureus* (29213)	4.00	0.05	0.60	0.40	0.01	166.70	250.00
*Staphylococcus aureus* (43300)	2.00	3.12	0.30	0.20	0.01	333.30	500.00
*Staphylococcus aureus (6538P)*	2.00	0.05	1.25	0.40	0.02	80.00	250.00
**Gram** −							
*Acinetobacter baumanni* (19606)	16.25	12.50	1.25	3.12	0.04	80.00	32.00
*Escherichia coli* (25922)	32.50	6.25	2.5	0.80	0.03	40.00	128.20
*Escherichia coli* (35218)	32.50	6.25	5.02	0.20	0.05	19.90	526.30
*Klebsiella pneumoniae* (13883)	4.00	12.50	5.00	0.20	0.05	20.00	500.00
*Pseudomonas aeruginosa* (27853)	32.5	6.25	2.5	0.20	0.03	40.00	526.30
**Clinical isolation ^a^**							
**Gram +**							
*Enterococcus faecalis* 1011	16.25	0.10	5. 00	0.20	0.05	20.00	500.00
*Enterococcus faecalis* 2011	16.25	0.10	5. 00	0.20	0.05	20.00	500.00
*Enterococcus faecalis* BS	16.25	0.40	5. 00	0.20	0.05	20.00	500.00
*Staphylococcus aureus* DEL	32.50	0.02	2.50	0.20	0.03	40.00	500.00
*Staphylococcus aureus* 23	32.50	0.40	0.30	0.80	0.01	333.30	125.00
*Staphylococcus aureus* 24	32.50	0.40	0.30	0.80	0.01	333.30	125.00
*Staphylococcus aureus* IAC	32.50	0.05	2.50	0.20	0.03	40.00	500.00
*Staphylococcus aureus* TER	32.50	0.05	2.50	0.40	0.03	40.00	250.00
**Gram** −							
*Acinetobacter baumanni* BS	65. 00	25. 00	0.60	0.20	0.01	166.70	526.30
*Citrobacter freundii* BS	32.50	12.50	2.50	0.20	0.03	40. 00	526.30
*Escherichia coli* ESBL	32.50	6.25	5. 00	0.20	0.05	20. 00	526.30
*Klebsiella pneumoniae* BS	65. 00	25. 00	0.20	1.25	0.01	500.00	80. 00
*Proteus mirabilis* BS	16.25	6.25	2.50	0.20	0.03	40.00	526.30

^a^ The acronyms refer to the disease of the patients from which the microorganisms were taken. Rfx, Rifaximine; FICI, Fractional Inhibitory Concentration Index.

## Data Availability

Not applicable.

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
