# Peer review of "Polyphenols from Olive-Mill Wastewater and Biological Activity: Focus on Irritable Bowel Syndrome†"

_nutrients, 2022, doi:10.3390/nu14061264_

Round 1

Reviewer 1 Report

Overall, this is an interesting paper that describes the action of polyphenols modulating the spontaneous and induced contractility of the ileum and the colon. The authors evaluated

MOMAST (Plus30, PW25, and HY100) on the contractility of the Guinea-pig -ileum, and colon. Also, it was evaluated the antimicrobial activity of these compounds.

They found that MOMASTs show a spasmolytic effect and antimicrobial activity.

Although the results presented in the manuscript might influence the field, there are several key errors in the manuscript and experimental design which lessen the potential impact this study will have on the field.

With the idea to help to improve the impact of the findings reported here I suggest taking into consideration the next,

  1. My main concern about this manuscript is that several parts are speculative. For example, the conclusion of this research is that they confirm that Plus30 could be used as a food supplement in patients with IBS (lines 33-24). Despite Plus30 showed a decrease in ileum and colon contractibility, and antimicrobial activity, this data was gotten using an ex-vivo preclinical study done in a healthy ileum-colon. I do not think the findings support the conclusion. It is necessary to rephrase the conclusion and add that is necessary for more research (i.e. toxicologic) to propose Plus30 as a food supplement to treat symptoms of IBS. We need to remember that FDA does not review food supplement products for safety and effectiveness before they are marketed.
  2. Figure 2A and B and C are very hard to interpret. I think the x and y-axis are wrong. In the text, it was said that papaverine was tested in a range of 0.01-10mM, but these concentrations do not correspond to what is shown in figure 2C. Why high concentration of papaverine test show no effect on contraction? The same question for Momast.
  3. The discussion is rather long and is rather speculative in places. Condensing and focusing the discussion would be beneficial.
  4. In lines 481-482 is said “ Plus30 in fact reduces slightly the tone with consequent effects on peristalsis and regulates the progress of the food bolus and promotes the absorption of nutrients”. I suggest rephrasing the sentence. No evidence of Plus30 on peristalsis, the progress of the food, and absorption of nutrients are shown in this research.
  5. Very speculative this sentence line 487-488 “For HY100, the increase in tone is linked to an increase in the medium frequencies which could be reflected in an increase in the speed of transit, but also in spasms related to pain”. Please rephrase.
  6. The final conclusion- section in the manuscript is rather long and speculative. The conclusion should be clear and concise.
  7. Indicate whether the data analyzed with parametric statistics follow a Gaussian distribution.
  8. It would be helpful to add asterisks to the figures when there are statistical differences.

Minor concerns

Line 574, double dot

Author Response

                                                                                                                   Bologna, 28 February 2022

Revision of Manuscript ID: nutrients-1599659

“Polyphenols From Olive Mill Wastewater & Biological Activity: Focus on Irritable Bowel Syndrome”

by: Francesca Curci , Filomena Corbo, Maria Lisa Clodoveo, Lara Salvagno, Antonio Rosato, Ivan Corazza, Roberta Budriesi, Matteo Micucci, and Laura Beatrice Mattioli

Dear Editor,

We have carefully read the comments of the Reviewers and we are grateful for the useful criticisms that allowed us to improve the quality of this study. The manuscript has been extensively revised, addressing all the points raised by the referees. Below we report a point-to-point reply to each specific criticism raised. We hope that in the present form it is suitable for publication.

Thank you again for your attention and we look forward to hearing from you soon.

                                                                                                        Sincerely

                                                                                     Roberta Budriesi Prof.

Reviewer 1:

Comments and Suggestions for Authors

Overall, this is an interesting paper that describes the action of polyphenols modulating the spontaneous and induced contractility of the ileum and the colon. The authors evaluated MOMAST (Plus30, PW25, and HY100) on the contractility of the Guinea-pig -ileum, and colon. Also, it was evaluated the antimicrobial activity of these compounds.

They found that MOMASTs show a spasmolytic effect and antimicrobial activity.

Although the results presented in the manuscript might influence the field, there are several key errors in the manuscript and experimental design which lessen the potential impact this study will have on the field.

With the idea to help to improve the impact of the findings reported here I suggest taking into consideration the next,

1. My main concern about this manuscript is that several parts are speculative. For example, the conclusion of this research is that they confirm that Plus30 could be used as a food supplement in patients with IBS (lines 33-24). Despite Plus30 showed a decrease in ileum and colon contractibility, and antimicrobial activity, this data was gotten using an ex-vivo preclinical study done in a healthy ileum-colon. I do not think the findings support the conclusion. It is necessary to rephrase the conclusion and add that is necessary for more research (i.e. toxicologic) to propose Plus30 as a food supplement to treat symptoms of IBS. We need to remember that FDA does not review food supplement products for safety and effectiveness before they are marketed.

We believe that the targets studied and modulated by the momasts are in agreement with the conventional drugs used in the therapy of IBS. However, we agree with the referee that it is appropriate to deepen the characterization to consider this a "food supplement" in accordance with the process of drug discovery, it is appropriate to continue with "food supplement discovery".

For this reason we have reframed both the abstract and the conclusions.

2. Figure 2A and B and C are very hard to interpret. I think the x and y-axis are wrong. In the text, it was said that papaverine was tested in a range of 0.01-10mM, but these concentrations do not correspond to what is shown in figure 2C. Why high concentration of papaverine test show no effect on contraction? The same question for Momast.

The figure has been corrected. The scale of panels A and B modified. In panel C the independent variable is expressed in M. In the paper (line 371) the unit of measurement has been corrected.

3. The discussion is rather long and is rather speculative in places. Condensing and focusing the discussion would be beneficial.

We agree with the referee that the discussion is quite long but unfortunately this derives from the numerous targets considered and from the need to link the results together and stress how MOMASTs are able to modulate more nodes of the target network of the considered pathology. In our opinion, it is therefore difficult to reduce the discussion without losing information.

4. In lines 481-482 is said “ Plus30 in fact reduces slightly the tone with consequent effects on peristalsis and regulates the progress of the food bolus and promotes the absorption of nutrients”. I suggest rephrasing the sentence. No evidence of Plus30 on peristalsis, the progress of the food, and absorption of nutrients are shown in this research.

The sentence has been changed as required.

5. Very speculative this sentence line 487-488 “For HY100, the increase in tone is linked to an increase in the medium frequencies which could be reflected in an increase in the speed of transit, but also in spasms related to pain”. Please rephrase.

The sentence has been rephrased.

6. The final conclusion- section in the manuscript is rather long and speculative. The conclusion should be clear and concise.

The final conclusion has been modified.

7. Indicate whether the data analyzed with parametric statistics follow a Gaussian distribution.

For what it concerns the analysis of spontaneous contractility, the Gaussian distribution was assessed through a Kolgomorov-Smirnov Test. We added the information to the paper.

8. It would be helpful to add asterisks to the figures when there are statistical differences.

Considering the large number of comparisons, only the non-significant differences have been added in Figures 4 to 7. The caption has been changed consistently.

Minor concerns

Line 574, double dot.

The additional dot has been removed.

Reviewer 2 Report

I read the manuscript, Polyphenols From Olive Mill Wastewater & Biological Activity: 2 Focus on Irritable Bowel Syndrome, with great interest and believe that is a well-written manuscript with a clear explanation of methods and results.

The authors examined three products, olive oil wastewater, named: MOMAST® (Plus30, PW25, and HY100). Based on the chemical composition, obtained with different methods, they hypothesized a possible application as food supplements in Irritable Bowel Syndrome. The results highlighted the ability of Plus30 to modulate spontaneous and induced contractility, exert a good antioxidant action, and significantly act on various microorganisms. These effects were found to be synergistic in the presence of antibiotics. In conclusion, they suggested that Plus30 could be used as a food supplement in patients with IBS.

In general, as I already mentioned above I think the manuscript is well written and certainly of considerable interest. I have some minor revisions to suggest:

  • Starting from figure 4 the figures are not high-quality pixels and are hard to read it would be nice if authors can provide better figures
  • I believe there are so many figures and tables presented for one paper which makes it hard for the reader to follow, I would recommend authors to compact their results and maybe according to the priority and main message put some of the figures and tables as an additional file - But this decision is up to authors since it is their effort and work-
  • In the statistical analysis part, the authors explain the tests but I could not see any statistical test or p values for tables and figures. the tests should be mentioned under the tables and the significance also should take part in figures and tables.
  • The authors presented their results with SEM but I believe here they are required to use SD. SD and SEM estimate quite different things. But in many articles, SEM and SD are used interchangeably and authors summarize their data with SEM as it makes data seem less variable and more representative. However, unlike SD which quantifies the variability, SEM quantifies uncertainty in the estimate of the mean. As readers here I think we are interested in knowing the variability within the sample and not the proximity of mean to the population mean, data should be precisely summarized with SD and not with SEM.  I would recommend they update their results with SD values. 
  • I would recommend authors define limitations and methodological weaknesses clearly since they recommend this extract as a treatment.

Author Response

                                                                                                                   Bologna, 28 February 2022

Revision of Manuscript ID: nutrients-1599659

“Polyphenols From Olive Mill Wastewater & Biological Activity: Focus on Irritable Bowel Syndrome”

by: Francesca Curci , Filomena Corbo, Maria Lisa Clodoveo, Lara Salvagno, Antonio Rosato, Ivan Corazza, Roberta Budriesi, Matteo Micucci, and Laura Beatrice Mattioli

Dear Editor,

We have carefully read the comments of the Reviewers and we are grateful for the useful criticisms that allowed us to improve the quality of this study. The manuscript has been extensively revised, addressing all the points raised by the referees. Below we report a point-to-point reply to each specific criticism raised. We hope that in the present form it is suitable for publication.

Thank you again for your attention and we look forward to hearing from you soon.

                                                                                                        Sincerely

                                                                                     Roberta Budriesi Prof.

Reviewer 2:

Comments and Suggestions for Authors

I read the manuscript, Polyphenols From Olive Mill Wastewater & Biological Activity: 2 Focus on Irritable Bowel Syndrome, with great interest and believe that is a well-written manuscript with a clear explanation of methods and results.

The authors examined three products, olive oil wastewater, named: MOMAST® (Plus30, PW25, and HY100). Based on the chemical composition, obtained with different methods, they hypothesized a possible application as food supplements in Irritable Bowel Syndrome. The results highlighted the ability of Plus30 to modulate spontaneous and induced contractility, exert a good antioxidant action, and significantly act on various microorganisms. These effects were found to be synergistic in the presence of antibiotics. In conclusion, they suggested that Plus30 could be used as a food supplement in patients with IBS.

In general, as I already mentioned above I think the manuscript is well written and certainly of considerable interest. I have some minor revisions to suggest:

- Starting from figure 4 the figures are not high-quality pixels and are hard to read it would be nice if authors can provide better figures.

The figures have been redone to improve their quality and modified as required. We consider this aspect related to spontaneous contractility to be particularly interesting for a potential application in integrated therapy. For this reason we believe it appropriate to leave the figures in the paper but we have added a summary table to make it easier to use the results obtained.

I believe there are so many figures and tables presented for one paper which makes it hard for the reader to follow, I would recommend authors to compact their results and maybe according to the priority and main message put some of the figures and tables as an additional file - But this decision is up to authors since it is their effort and work.

We understand the reader's difficulty in following the results relating to spontaneous motility. For this reason we have added a summary table of the effects of the MOMASTs on the parameters considered.

- In the statistical analysis part, the authors explain the tests but I could not see any statistical test or p values for tables and figures. the tests should be mentioned under the tables and the significance also should take part in figures and tables.

Statistics have been added in all tables and figures.

- The authors presented their results with SEM but I believe here they are required to use SD. SD and SEM estimate quite different things. But in many articles, SEM and SD are used interchangeably and authors summarize their data with SEM as it makes data seem less variable and more representative. However, unlike SD which quantifies the variability, SEM quantifies uncertainty in the estimate of the mean. As readers here I think we are interested in knowing the variability within the sample and not the proximity of mean to the population mean, data should be precisely summarized with SD and not with SEM. I would recommend they update their results with SD values.

All data were re-analyzed, reporting SD.

I would recommend authors define limitations and methodological weaknesses clearly since they recommend this extract as a treatment.

As suggested, we have rewritten both abstracts and conclusions

Round 2

Reviewer 1 Report

Thanks to the authors for doing my suggestions. However, I still think Figure 2 is hard to understand. MOMASTs were tested in high K+ induce-contraction in both ileum (Fig. 2A) and colon (Fig 2B), but the legend for the y axis is different between 2A and 2B. 
Also, I suggest Adding how was calculated the % of contraction shown in figure 2.

Linea 203, says that was tested Paparevine in a range of 0.01-10 mM, but in figure 3 the authors tested lower concentrations.

Papaverine was used as a control positive which inhibits smooth muscle contraction. As I interpret from figure 2C the height concentration of papaverine show 100% of contraction. Why this contradiction. The legend of the y-axis is wrong?
This contradiction is present in Figure 2B. Why does the highest concentration of MOMASTs produce 100% of contraction? 

Author Response

                                                                                                                   Bologna, 09 March 2022

Revision 2 of Manuscript ID: nutrients-1599659

“Polyphenols From Olive Mill Wastewater & Biological Activity: Focus on Irritable Bowel Syndrome”

by: Francesca Curci , Filomena Corbo, Maria Lisa Clodoveo, Lara Salvagno, Antonio Rosato, Ivan Corazza, Roberta Budriesi, Matteo Micucci, and Laura Beatrice Mattioli

Dear Editor,

We have carefully read the comments of the Reviewer and we are doing all the suggestions. The manuscript has been revised and all mistakes corrected. Below we report a point-to-point reply to each specific criticism raised. We hope that in the present form it is suitable for publication.

Thank you again for your attention and we look forward to hearing from you soon.

                                                                                                         Sincerely

                                                                                     Roberta Budriesi Prof.

Reviewer 1:

Comments and Suggestions for Authors

Thanks to the authors for doing my suggestions. However, I still think Figure 2 is hard to understand.

We thank the referee for his patience as some inaccuracies have emerged for which we apologize. We have corrected all errors as explained below.

MOMASTs were tested in high K+ induce-contraction in both ileum (Fig. 2A) and colon (Fig 2B), but the legend for the y axis is different between 2A and 2B.

Thanks for seeing that the caption in the y-axis of figures 2b and 2c was wrong. We have proceeded to correct. MOMAST and PAPAVERINA cause inhibition of potassium-induced contraction.

Also, I suggest Adding how was calculated the % of contraction shown in figure 2.

We apologize but this is an inhibition calculated as a % inhibition of the contraction induced by potassium (see materials and methods).

Linea 203, says that was tested Papaverine in a range of 0.01-10 mM, but in figure 3 the authors tested lower concentrations.

The error in line 203 has been corrected.

Papaverine was used as a control positive which inhibits smooth muscle contraction. As I interpret from figure 2C the height concentration of papaverine show 100% of contraction. Why this contradiction. The legend of the y-axis is wrong?

The legend of the y-axis is wrong and has been correct.

This contradiction is present in Figure 2B. Why does the highest concentration of MOMASTs produce 100% of contraction?

As mentioned above, some panels of figure 2 contains some errors that have been detected by the referee and have been corrected.
